# Nutritional Quality and Overall Acceptability Optimization of Ultra-Fast Air-Superchilled Golden Rainbow Trout (*Oncorhynchus mykiss*, Stevanovski) Using the Response Surface Methodology



Vladimir D. Kitanovski [1], Stefan G. Dragoev [2,*], Hristo N. Nikolov [3], Desislava B. Vlahova-Vangelova [2] and Dessislav K. Balev [2]

[1]  Department of Food Technology, Faculty of Technological Sciences, University Mother Teresa, Skopje, 4 Mirche Acev Str., 1000 Skopje, North Macedonia; vladimir.kitanovski@unt.edu.mk
[2]  Department of Meat and Fish Technology, Technological Faculty, University of Food Technologies, 26 Maritza Blvd., 4002 Plovdiv, Bulgaria; d_vangelova@uft-plovdiv.bg (D.B.V.-V.); d_balev@uft-plovdiv.bg (D.K.B.)
[3]  Department of Preservation and Refrigeration Technology, Technological Faculty, University of Food Technologies, 26 Maritza Blvd., 4002 Plovdiv, Bulgaria; nikar@abv.bg
*  Correspondence: s_dragoev@uft-plovdiv.bg; Tel.: +359-32-603-802 or +359-893-387-647 or +359-899-829-920

**Featured Application: The findings can be implicated by rainbow trout farmers and are directly influenced by fish shelf life in supermarkets. The findings will be interesting for consumers, too.**

**Abstract:** Temperatures below the cryoscopic point help to partially freeze most of the water in the fish muscle tissue. This reduces water activity and makes the remaining free water hardly accessible to microorganisms. The objective of this study was to determine the best process regime of ultra-fast air-superchilling, giving us the optimal quality of golden rainbow trout. Two hundred and thirty-four live golden rainbow trout (*Oncorchynchus mykiss*, Stevanovski) (18 groups of 13 fish in a group) were caught and immediately stunned by an electric current (P = 42 W). The stunned fish was placed in styrofoam cans and covered with flaked ice. The sensory analysis, total nitrogen volatile bases (TVB-N), total number of microorganisms (TVC), and presence of biogenic amines were determined. According to the optimized values for TVB-N, TVC, and sensory scores, giving us a better quality of ultra-fast air-superchilled golden rainbow, the process regime has been found at the following parameters: air temperature T = −11.3 °C; airflow velocity υ = 6.5 m s$^{-1}$; and packaging layer thickness D = 79.2 μ. The superchilled golden rainbow trout processed by this regime has the lowest degree of proteolytic degradation, delayed development of the microflora, and retains the best possible sensory properties and freshness.

**Keywords:** fish; freshness; shelf life; coolant velocity; coolant temperature; package thickness; total volatile base nitrogen (TVB-N); total number of mesophilic microorganisms (TVC); histamine; sensory properties

## 1. Introduction

Albinism in the golden rainbow trout is a manifestation of the characteristic recessive gene that suppresses the activity of tyrosinase. It is due to a specificity in albinism genes such as Tyr-1, Tyr-2, and slc45a2 associated with pigmentation [1]. The selective breeding of the albino trout aquaculture achieves good yield, accompanied by the golden color of the fish, making it attractive to the consumer. Recent evidence suggests that temperature plays a key role in rainbow trout development and can significantly affect survival and compromise fish shelf life [2]. The specific genetic characteristic of the golden rainbow trout affects its proximate composition [3] and, in particular, the fatty acid composition

of endogenous lipids. The golden rainbow trout is characterized by a higher content of muscle lipids and omega-3 polyunsaturated fatty acids. This leads to a lower oxidative lipid stability in the albino form of the fish during refrigeration compared to the normal genotypic form [4]. The specific flavor, color, texture, and nutritional value of the golden rainbow trout are significantly affected by susceptibility to the course of phospholipid oxidation in muscle tissue and skin [5]. Fresh golden rainbow trout is most commonly available on the market in a chilled or frozen state. Unfortunately, cooling and especially freezing change the proximal composition, physicochemical characteristics, microbial status, and sensory properties of the fish [6], which adversely affects its nutritional value and general quality [7]. Parallel to this situation, quantity, services, environmental effects, and cost play a major role in the fish industry [8]. In the last years, thanks to the dietary advice for a healthy lifestyle, where fish products represent an essential component of a balanced diet, consumption has seen a huge demand increase [9]. Logically, the amount of fish waste was noticed to have a huge increase, too. Therefore, the fish waste management topic became very popular because it can decrease the impacts of fishery discards on the environment and help the food value chain not to lose quality grades of fish products and prevent unnecessary processing [10]. Nowadays, fish waste processing is of low profitability, with the production of fish oil, fish meal, or raw material for feeding in aquaculture [11,12]. Based on this, we can act in two ways to improve the fish waste management system and to develop high-value processed ingredients or to develop techniques that can deliver quality fish products to consumers with extended shelf life. On the other hand, it is necessary to look for new innovative approaches to the technology of fish refrigeration [13], which helps preserve high quality for as long as possible with the best carbon footprint ratio. Superchilling of food products comes as an alternative to offering high-quality fresh fish on the market with significantly extended shelf life [14]. In this sense, the temperature is a major technological factor that determines the shelf life of fish [15]. Bao et al. [16] have established significant advantages in storing Arctic trout (*Salvelinus alpinus*) at $-2\,°C$ and reported the prolongation of shelf life of up to six days compared to cold-stored fish at $+3\,°C$. Nevertheless, fish superchilling may result in partial freezing. It may reduce the water-holding capacity of the fish meat and release the drip loss. However, compared to freezing and cooling, superchilling has many advantages in preserving fish quality [17,18]. The degree of muscle protein denaturation and the rate of tissue structural damage in superchilled fish is lower compared to the frozen one [19]. Garimela and Schroeder [20] have conducted a theoretical analysis of the local heat transfer distributions in confined multiple air jet impingement and have proven the benefits of this method of cool processing for foods such as trout. Sarkar et al. [21] have studied fluid streams and heat transfer under air-jet impingement during food processing. Nevertheless, the mechanisms, techniques, and methods of superchilling of fresh fish have not yet been thoroughly studied. Studies need to be conducted to develop an effective strategy to inhibit the hydrolytic and oxidative changes in fish during superchilling and further during future refrigerated storage. Our earlier studies have demonstrated the possibility of the shelf-life prolongation of the golden rainbow trout by ultra-fast air or cryogenic $CO_2$ superchilling [22]. Analyses showed that ultra-fast air-processed samples had better quality and safety grades, but optimization of the superchilling system variables was needed, so the emphasis is placed on the established advantages of the ultra-fast air-superchilling method. That is why the objective of this study was to determine the best process regime of ultra-fast air-superchilling, giving us the optimal quality of golden rainbow trout. To reach this study's aim, we employed a Hybrid Chaotic Big Bang-Big Crunch (HCBB-BC) mapping algorithm for improving the performance of three-dimensional Network-on-Chip (3D NoC) systems. Erikson et al. [23] suggest the use of NMR to find the relationship between optimal storage temperature and flesh quality of Atlantic salmon, measuring the echo time, relaxation delay, number of scans, and number of echoes at three targeted ice fractions. Optimization of the superchilling process of golden rainbow trout is a comprehensive exploration of a topic that holds

significant relevance in production, carbon footprint, supply chain, and consumer trust and satisfaction level.

## 2. Materials and Methods

### 2.1. Flow Chart of the Optimization Process of Golden Rainbow Trout Air-Jet Superchilling Regime

Two hundred and thirty-four live golden rainbow trout (*Oncorchynchus mykiss*, Stevanovski) with a live weight of 370 ± 10 g each [18 groups × 13 fish in a group] were purchased from the fish farm Ribena Ltd. (Zlatna Panega village, Yablanitsa municipality, Lovech region, Bulgaria). Immediately after the catch, the live fish was stunned by electric current power (P = 42 W). Later, such electrified fish were placed in refrigerated cans layered with styrofoam and covered with flaked ice. Within a one-hour period, the fish was delivered to the laboratories of the Department of Meat and Fish Technology at the University of Food Technologies, Plovdiv. The trout was gutted as quickly as possible. The sub-spinal blood vessel and the black abdomen membrane were also removed. The design of the experimental work was made according to the flow chart presented in Figure 1 and Supplementary Materials of the apparatures for vacuum-packing and superchilling can be found in Supplementary Materials. Thus, the gutted fish were divided into 18 groups, consisting of 13 fish in each group. From each group, five fish were used for sensory assessment with one complex score, taste after heat treatment, and overall acceptability obtained by the symbiosis of more analyzed sensory indicators. From the remaining 8 fish, average laboratory samples were prepared and used to determine the total nitrogen volatile bases (TVB-N), the total number of microorganisms (TVC), and the presence of biogenic amines. The first group of 13 fish was analyzed 2 h post mortem. This group of fish was used as a control sample. The results obtained were used as baseline levels (0 d of storage) against which the degree of change in the superchilled golden rainbow trout was evaluated. The remaining 221 golden rainbow trout (17 groups of 13 fish) were individually packed in polyvinyl dichloride (PVDC) zipped bags and were superchilled according to the regimes shown in Table 1.

**Table 1.** Matrix of the exploratory factor analysis—experiment with three factors at five levels designed by using Central Composite Design (CCD).

|  | X1 | X2 | X3 | (T, °C) | ($v$, m s$^{-1}$) | (D, μm) |
|---|---|---|---|---|---|---|
| 1 | − | − | − | −20.0 | 3.0 | 60.0 |
| 2 | + | − | − | −5.0 | 3.0 | 60.0 |
| 3 | − | + | − | −20.0 | 7.0 | 60.0 |
| 4 | + | + | − | −5.0 | 7.0 | 60.0 |
| 5 | − | − | + | −20.0 | 3.0 | 100.0 |
| 6 | + | − | + | −5.0 | 3.0 | 100.0 |
| 7 | − | + | + | −20.0 | 7.0 | 100.0 |
| 8 | + | + | + | −5.0 | 7.0 | 100.0 |
| 9 | −a | 0 | 0 | −25.0 | 5.0 | 80.0 |
| 10 | +a | 0 | 0 | 0 | 5.0 | 80.0 |
| 11 | 0 | −a | 0 | −12.5 | 1.6 | 80.0 |
| 12 | 0 | +a | 0 | −12.5 | 8.4 | 80.0 |
| 13 | 0 | 0 | −a | −12.5 | 5.0 | 46.4 |
| 14 | 0 | 0 | +a | −12.5 | 5.0 | 113.6 |
| 15 | 0 | 0 | 0 | −12.5 | 5.0 | 80.0 |
| 16 | 0 | 0 | 0 | −12.5 | 5.0 | 80.0 |
| 17 | 0 | 0 | 0 | −12.5 | 5.0 | 80.0 |

X1, X2, and X3 are factorial levels in the experimental design, and T = air temperature; $v$ = airflow velocity, and D = package thickness are variable factors.

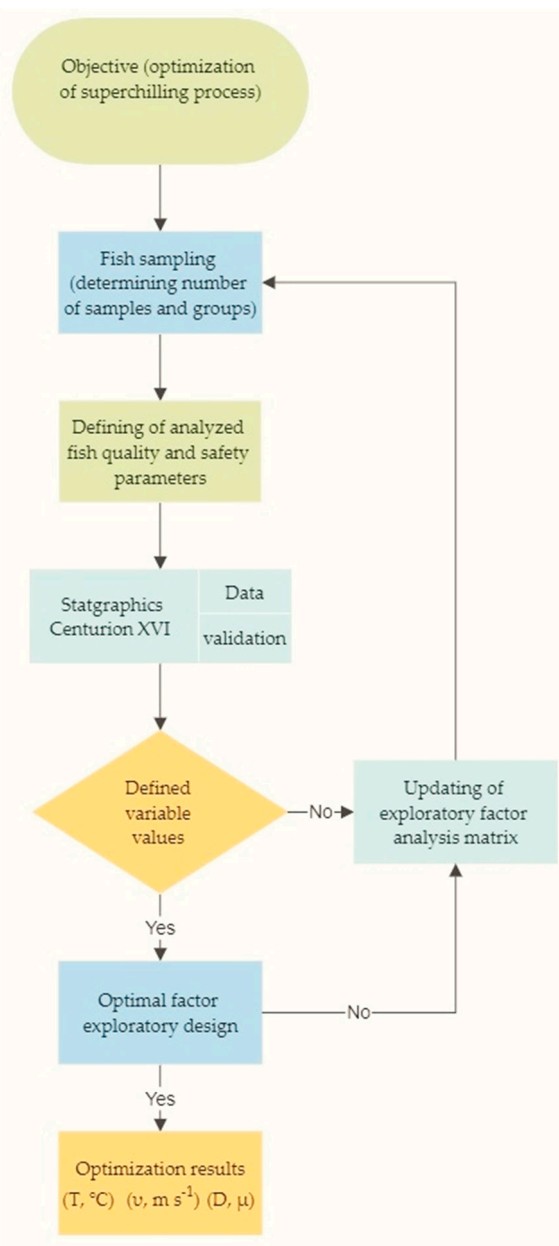

**Figure 1.** A flowchart of the optimization of the process of golden rainbow trout air-jet superchilling.

### 2.2. Auxiliary Materials

The following auxiliary materials were used in the experiments: flaked ice, purchased from Ribena Ltd. (Zlatna Panega village, Yablanitsa municipality, Lovech region, Bulgaria); styrofoam refrigeration cans, supplied by Styropor BG-1 Ltd., Stara Zagora, Bulgaria. Their parameters were as follows: material—EPS-F (expanded polystyrene foam, styrofoam), internal dimensions 525/390/40 mm; the wall thickness is 4 cm, heat conductivity coefficient 0.034–0.038 W m$^{-1}$ K$^{-1}$, color—white with grainy structure, mass of the material of 15–40 kg m$^{-3}$, granule content—98% air and 2% polystyrene, hygroscopicity of 0.05–0.20%, water absorption was no more than 2–3% of the volume. Moisture and gas impermeable polyvinyl/dichloride PVDC bags for individual fish packaging in a vacuum, with a thickness of 60, 80, and 100 μm, respectively, were produced and donated by Intrama Bulgaria Ltd., Plovdiv, Bulgaria.

### 2.3. Exploratory Factor Analyses Design

The optimization process of ultra-fast air-superchilling of golden rainbow trout (*Oncorhynchus mykiss*, Stevanovski) was based on an exploratory factor analysis design [24], analyzing three factors on five levels (Table 1). The design was organized on the basis of the results obtained from previous studies [22]. Three factors were analyzed on two levels, upper factor level (+) and lower factor level (−), maximum upper factor level (+a), and maximum lower factor level (−a), at three reps in the center (0). The selected model represents an exploratory factor analysis with 17 combinations, according to the recommendations of [25].

The independent variables (factors) include air temperature X1 (T, °C), airflow velocity X2 ($\upsilon$, m s$^{-1}$), and thickness of PVDC material from which the vacuum packing bags were produced X3 (D, μm). For determining the quality of dependent variables, we used the physico-chemical indicator for the freshness of fish—total volatile base nitrogen (TVB-N), the microbiological criterion for technological hygiene and safety—total number of mesophilic microorganisms (TVC), and the established complex sensory score of the sensory evaluation of golden rainbow trout after heat treatment—a parameter which reflects the aromatic properties and the consistency of the roasted fish as the most important sensory characteristic. The matrix of the experiment is presented in Table 2.

**Table 2.** Matrix of the independent variables used in Central Composite Design (CCD).

| Independent Variables | Upper Factor Levels | Lower Factor Levels | Centre of the EFA Design | Minimal Upper Factor Level | Maximal Lower Factor Level | Intervals of Variations |
|---|---|---|---|---|---|---|
| Temperature X1 (T, °C) | −5.0 | −20.0 | −12.5 | −25.0 | 0.0 | ± 7.5 |
| Airflow velocity X2 ($\upsilon$, m s$^{-1}$) | 7.0 | 3.0 | 5.0 | 1.6 | 8.4 | ± 2.0 |
| Package thickness X3 (D, μm) | 100.0 | 60.0 | 80.0 | 46.4 | 113.6 | ± 20.0 |

Defined upper, lower, and central level, and their minimal and maximal values for all experimental analyses' runs.

The maximum upper and lower levels of the independent variables were set following the recommendations of [26]. All 17 experimental groups of golden rainbow trout were individually vacuum packed in PVDC envelope bags and processed according to the corresponding quality indicators at 21 d of the storage from −2 to −3 °C. On the basis of these three response functions, the ultra-fast air-superchilling process of golden rainbow trout was optimized. In addition, the content of the biogenic amines in fish flash was investigated as the regulated safety indicator at the end of the storage (21 d) at a temperature from −2 to −3 °C.

### 2.4. Sensory Analyses

A sensory analysis was carried out by a five-member panel of trained experts. The analyses were conducted at the scientific laboratory of the Department of Meat and Fish Technology at the University of Food Technologies, Plovdiv, Bulgaria. For this purpose, we used a nine-grade scale [27] modified according to the specific needs of our experiments. According to this scale, grade 1 represents the extremely negative assessment of the analyzed indicator, while grade 9 is the greatest possible satisfaction. The complex evaluation unit for the fish sensory properties was evaluated immediately after the fish had been roasted on an electric grill. On the basis of the obtained sensory score by each of the five committee members, we have calculated an arithmetic mean.

### 2.5. Determination of Volatile Base Nitrogen (TVB-N)

This method determines the number of bound and free nitrogenous volatile bases such as ammonia, trimethylamine, dimethylamine, monomethylamine, etc., obtained by proteolytic degradation of fish meat proteins. Average laboratory samples were prepared of homogenized fish meat. Of each average laboratory sample with the amount of 1 g of fish muscle, homogenate was used. TVB-N were distilled with water vapor and then caught in an acidic solution with known concentration [28]. Their quantity is determined by colorimetric titration with Nessler's reagent, whereby the color of the sample changes from red to yellow at the equivalent point.

### 2.6. Determination of Total Viable Count (TVC) of the Aerobic Mesophilic Microorganisms

The TVC determination in fish muscle tissue was carried out by the method described by [29] after 72 h incubation at 28 °C.

### 2.7. Determination of Biogenic Amines

The biogenic amines in fish flash were determined by the HPLC method [30]. Five grams of trout meat were minced. A 10 $cm^3$ 0.4 M $HClO_4$ (perchloric acid) was added to the 5 g of minced fish. The resulting mixture was homogenized for 20 min with a shaker. The homogenate was centrifuged for 10 min at 6500 $min^{-1}$ and was filtered through filter paper. The obtained filtrate was collected in a 100 $cm^3$ Erlenmeyer flask and then subjected to derivatization. A 200 $mm^3$ filtrate was replaced in an Eppendorf test tube. Forty $mm^3$ of 2 N NaOH, 60 $mm^3$ of saturated sodium bicarbonate ($NaHCO_3$), 20 $mm^3$ internal standard, and 380 $mm^3$ of diethyl chloride was added. The mixture was allowed to rest for 15 min in the dark, and then 10 $mm^3$ of 25% $NH_4OH$ was added. After 5 min, the resulting composition was filtered through a syringe filter and injected into the HPLC chromatograph column (Hitachi LaChrom Elite L-2300) with the following characteristics: column oven—temperature range, cooling: room temperature to −15 °C; temperature range, heating: room temperature to +50 °C; setting range: 1 °C to 65 °C; accuracy: ±1 °C; safety: built-in vapor and leak sensor; stability: ±0.1 °C; column capacity: up to three 25 cm columns; dimensions (W × H × D): 13.5 × 15.75 × 6 in; weight: ~22 lb, (10 kg).

### 2.8. Statistical Analyses

For optimization of the process of ultra-fast air-superchilling, we use the method of exploratory factor analysis [25]: experiment with three factors at five levels designed by using Central Composite Design (CCD). For this final purpose, a desirability test was performed, and the visualization was presented with a desirability plot. All statistical procedures were performed using the STATGRAPHICS Centurion XVI software version I.

## 3. Results

### 3.1. Optimization of the Ultra-Fast Air-Superchilling Process of Golden Rainbow Trout Referring to Quality and Safety Parameters

The influence of the three examined variables, the air temperature (T), the airflow velocity (υ), and the packing layer thickness (D) on the TVB-N formation, the total viable count (TVC), and the sensory established complex scores of grilled golden rainbow trout stored at superchilled temperatures are presented in Table 3. It was found that the lowest content of TVB-N (23.23 ± 0.82 mg $N_2$ 100 $g^{-1}$), the smallest total viable count of microorganisms (TVC = 4.3 log cfu $g^{-1}$), and the highest sensory score (8.50 ± 0.19) were obtained for sample 9 (air temperature = −25 °C; airflow velocity = 5 m $s^{-1}$ and packing layer thickness = 80 μm).

**Table 3.** The muscle tissue TVB-N content, total viable count of microorganisms, and complex sensory score of grilled ultra-fast air-superchilled golden rainbow trout (*Oncorhynchus mykiss*, Stevanovski) stored 21 days from −2 to −3 °C.

| No | T °C | υ m s$^{-1}$ | D μm | TVB-N mg N2 100 g$^{-1}$ | TVC log cfu g$^{-1}$ | Sensory Scores |
|---|---|---|---|---|---|---|
| 1 | −20.0 | 3.0 | 60.0 | 27.88 ± 1.12 [a] | 4.76 [e] | 7.25 ± 0.18 [g] |
| 2 | −5.0 | 3.0 | 60.0 | 25.83 ± 0.98 [b] | 4.92 [e] | 7.15 ± 0.13 [g] |
| 3 | −20.0 | 7.0 | 60.0 | 26.73 ± 1.22 [b] | 4.95 [e] | 8.00 ± 0.15 [g,h] |
| 4 | −5.0 | 7.0 | 60.0 | 24.76 ± 0.87 [c] | 5.01 [e] | 8.45 ± 0.19 [i] |
| 5 | −20.0 | 3.0 | 100.0 | 27.80 ± 1.14 [c,a] | 4.77 [e] | 7.00 ± 0.12 [g] |
| 6 | −5.0 | 3.0 | 100.0 | 25.82 ± 1.25 [c,b] | 4.94 [e] | 7.25 ± 0.11 [g] |
| 7 | −20.0 | 7.0 | 100.0 | 25.91 ± 0.98 [c,b] | 4.98 [e] | 7.75 ± 0.14 [g,h] |
| 8 | −5.0 | 7.0 | 100.0 | 25.17 ± 1.07 [c,b] | 4.57 [e] | 8.25 ± 0.17 [g,h] |
| 9 | −25.0 | 5.0 | 80.0 | 23.23 ± 0.82 [d] | 4.13 [f] | 8.50 ± 0.19 [i] |
| 10 | 0 | 5.0 | 80.0 | 29.88 ± 1.12 [c] | 5.72 [e] | 7.00 ± 0.18 [g] |
| 11 | −12.5 | 1.6 | 80.0 | 28.83 ± 0.98 [c] | 5.11 [e] | 7.25 ± 0.13 [g] |
| 12 | −12.5 | 8.4 | 80.0 | 27.73 ± 1.22 [b,c] | 4.81 [e] | 7.75± 0.15 [g,h] |
| 13 | −12.5 | 5.0 | 46.4 | 29.76 ± 0.87 [c] | 5.22 [e] | 8.00 ± 0.19 [g] |
| 14 | −12.5 | 5.0 | 113.6 | 26.82 ± 1.14 [b] | 4.93 [e] | 7.50 ± 0.12 [g] |
| 15 | −12.5 | 5.0 | 80.0 | 25.19 ± 1.25 [c] | 5.01 [e] | 7.75 ± 0.11 [g,h] |
| 16 | −12.5 | 5.0 | 80.0 | 25.28 ± 1.23 [c] | 4.89 [e] | 7.75 ± 0.11 [g,h] |
| 17 | −12.5 | 5.0 | 80.0 | 25.32 ± 1.21 [c] | 4.97 [e] | 7.75 ± 0.11 [g,h] |

Data are expressed as an average value ± standard deviation, (N = 5). Values with different letters in superscript within the same column differ significantly at $p < 0.05$. T = air temperature in the chamber; υ = airflow velocity; D = package thickness; TVB-N = total volatile base nitrogen; TVC = total viable counts.

*3.2. Influence of the Superchilling Process Variables on the Condition of Quality and Safety Parameters of Golden Rainbow Trout during Storage*

The impact and significance of variable factors, air temperature, packing layer thickness, and airflow velocity on the condition of quality and safety examined parameters during 21 days of storage of golden rainbow trout at sub-cryoscopic temperature intervals from −2 °C to −3 °C is presented in Figures 2 and 3. The standardized Pareto graphs testify that the air-flow velocity (υ) has the strongest influence on the sensory score and is slightly weaker on the TVB-N content and total viable count of microorganisms (TVC). The factor air temperature (T) has the second strongest influence in relation to TVB-N content and TVC, and the factor packing layer thickness (D) has a comparatively smaller but significant impact on those two parameters.

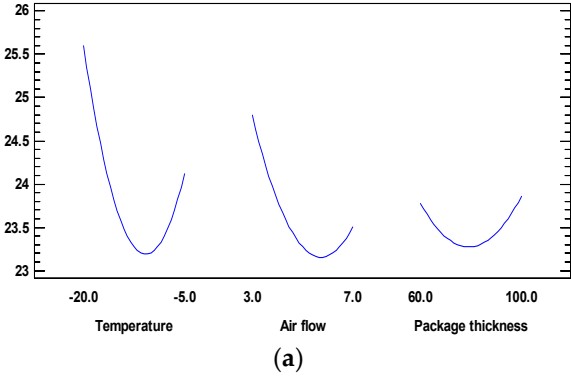

(a)

**Figure 2.** *Cont.*

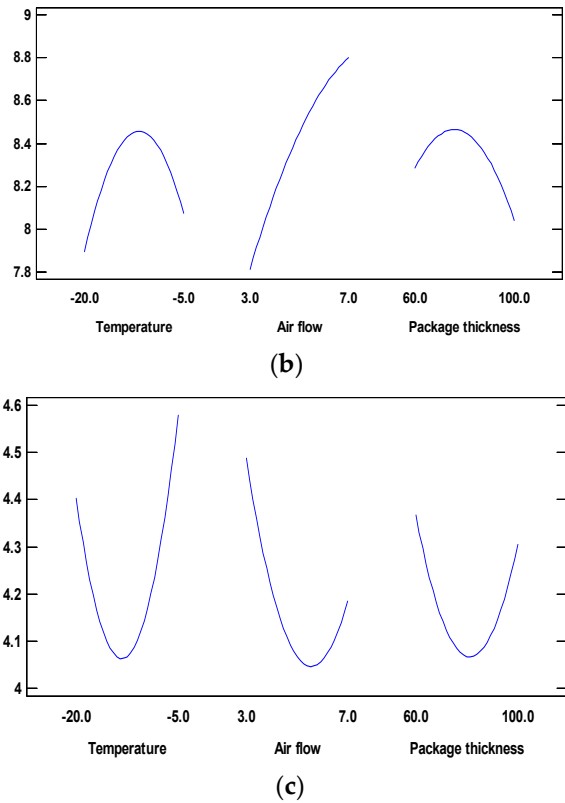

**Figure 2.** Influence of variables on quality and safety parameters condition of ultra-fast air super-chilled golden rainbow trout (*Oncorhynchus mykiss*, Stevanovski) stored for 21 days at temperatures from −2 to −3 °C; (**a**) Impact on TVB-N content; (**b**) Impact on total viable counts (TVC) formation; (**c**) Impact on established comlex sensory score of grilled fish.

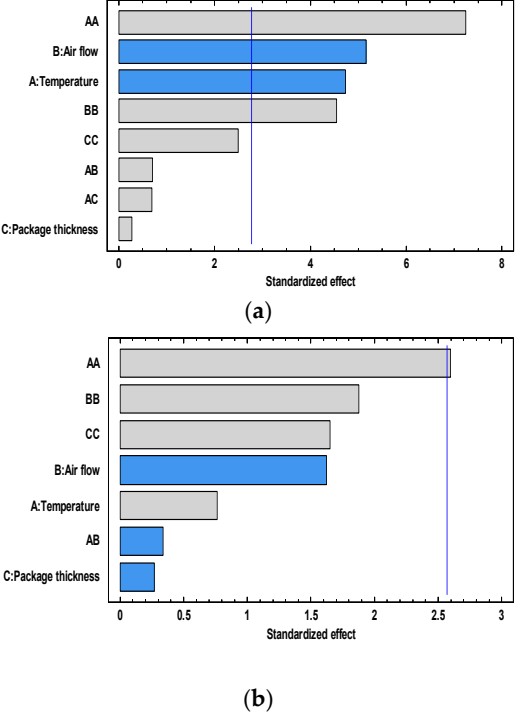

**Figure 3.** *Cont.*

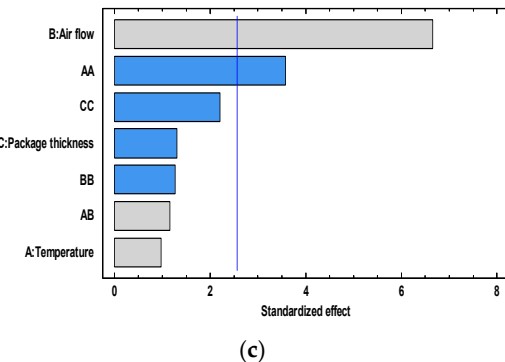

(**c**)

**Figure 3.** Significance of variables on quality and safety parameters condition of ultra-fast air super-chilled golden rainbow trout (*Oncorhynchus mykiss*, Stevanovski) stored for 21 days at temperatures from −2 to −3 °C; (**a**) Significance on TVB-N content in muscle tissue; (**b**) Significance on total viable counts (TVC) formation; (**c**) Significance on established complex sensory score of grilled fish.

### 3.3. Mathematical Optimization of the Ultra-Fast Air-Superchilling Process of Golden Rainbow Trout

Using the mathematical optimization method, the optimal ultra-fast air-superchilling regime of golden rainbow trout was established, thus providing the opportunity for the fish to be successfully stored for 21 days from −2 to −3 °C (Figure 4).

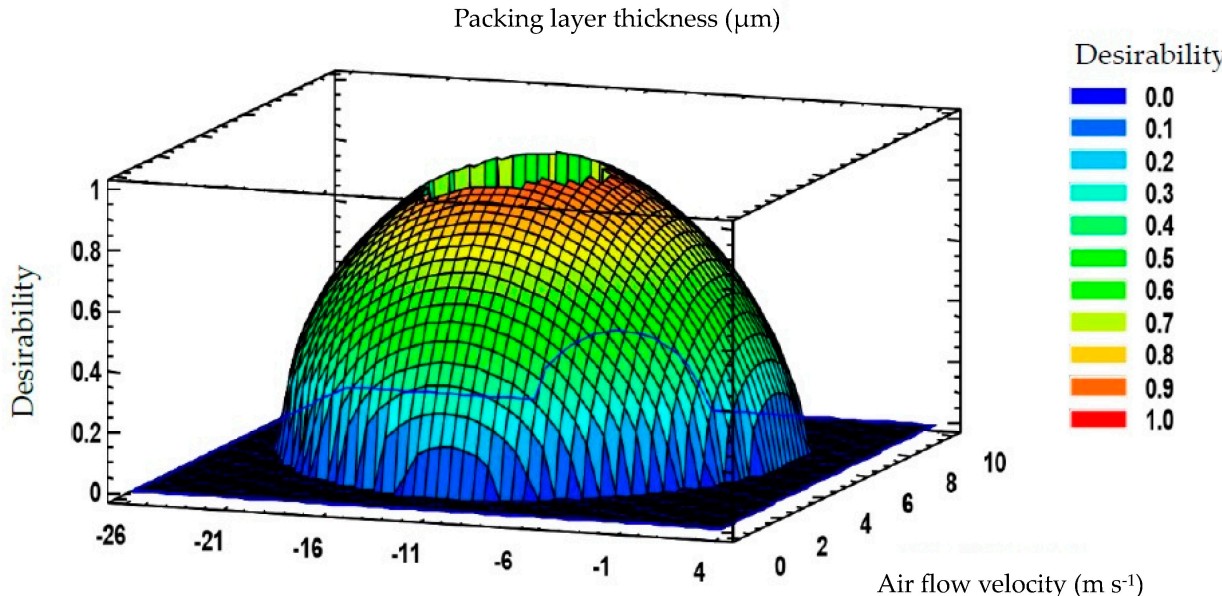

**Figure 4.** Graphical three-dimensional image of complex optimum of the two examined variables: the air temperature (T) and the airflow velocity (υ) at ultra-fast air-superchilling of the golden rainbow trout (*Oncorchynchus mykiss*, Stevanovski).

The selected exploratory factor analysis of the CCD model with 17 combinations provides optimized results for selected variables of the ultra-fast air superchilling process, presented in Figure 5 below.

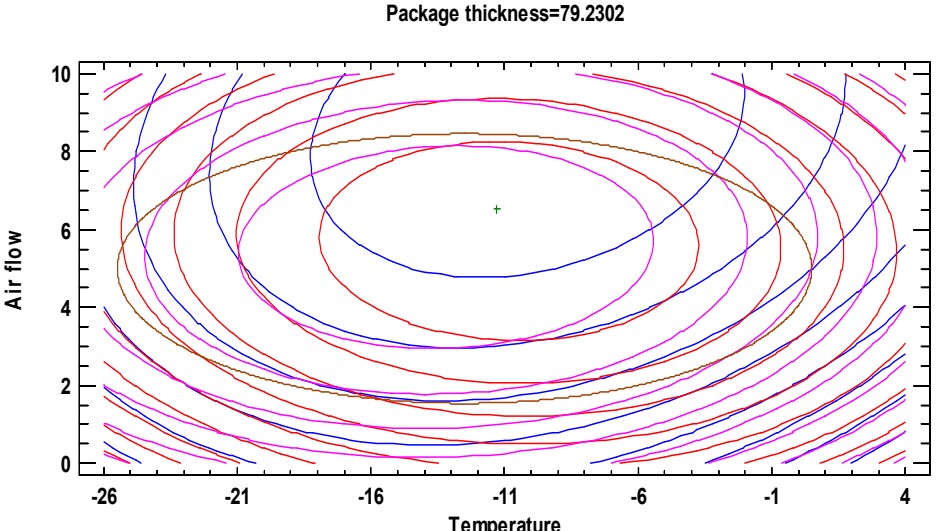

**Figure 5.** Overlay plot of optimized values for three examined variables: the air temperature (T); the airflow velocity (υ); and the packing layer thickness (D) at ultra-fast air-superchilling of the golden rainbow trout (*Oncorchynchus mykiss*, Stevanovski).

An optimal ultra-fast air-superchilling regime of the golden rainbow trout (*Oncorchynchus mykiss*, Stevanovski) has been found at the air temperature T = −11.3 °C, the airflow velocity υ = 6.5 m s$^{-1}$ and the packing layer thickness D = 79.2 μm.

*3.4. Presence of Biogenic Amines in Golden Rainbow Trout (Oncorhynchus mykiss, Stevanovski) Superchilled at Optimal Ultra-Fast Air Regime*

No presence of biogenic amines was detected in any of the samples tested in superchilled golden rainbow trout after 21 days of storage from −2 to −3 °C, which is evident from the HPLC chromatogram of sample 9 presented in Figure 6.

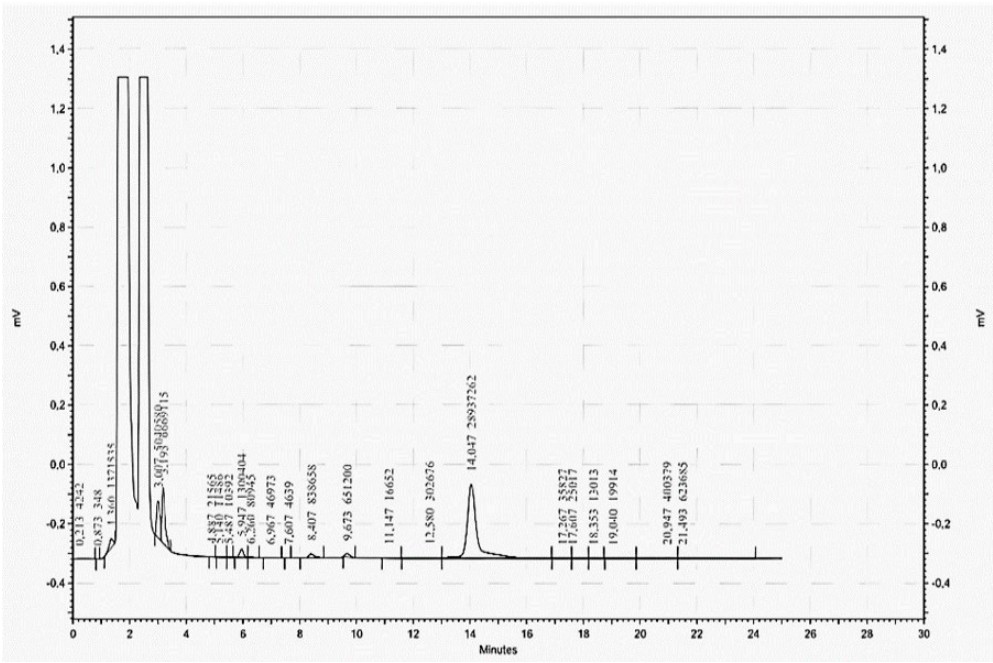

**Figure 6.** HPLC chromatogram proving the absence of biogenic amines in air jet impingement superchilled golden rainbow trout (*Oncorchynchus mykiss*, Stevanovski) after 21 days of storage from −2 to −3 °C.

## 4. Discussion

Our obtained results regarding the formation of TVB-N during storage from $-2$ to $-3$ °C can be explained by the effects of endogenous or bacterial proteolytic enzyme systems [31]. The TVB-N values in all 17 tested samples (Table 3) were very low and did not exceed the limits of acceptability during the 21 days of superchilled storage. This means that, at all regimes of superchilling, the fish preserved its freshness to a satisfactory degree. In accordance with the obtained results, [32] reported a possible shelf life of 21 days for frozen grass carp (*Ctenopharyngodon idellus*).

From the results presented in Table 3, it can be seen that the combination of factors in sample 9 (in the center of the experiment; $T = -12.5$ °C; $v = 5$ m s$^{-1}$; $D = 80$ μm) TVB-N were with the lowest values of 23.23 mg N$_2$ 100 g$^{-1}$, while the fish in sample 1 ($T = -20$ °C; $v = 3$ m s$^{-1}$; $D = 60$ μm) had the highest values of 27.88 mg N$_2$ 100 g$^{-1}$.

The influence of the three examined variables, the air temperature (T), the airflow velocity ($v$), and the packing layer thickness (D) to the total viable count (TVC) of golden rainbow trout during the 21 days of superchilled storage showed different values in all seventeen studied superchilling regimes. The lowest TVC was found in the superchilled trout—sample 9 = 4.13 log cfu g$^{-1}$, which is below the acceptable limit of 7.00 log cfu g$^{-1}$ [33]. It was established that only the factor of air temperature (T) significantly influences microflora growth (Figure 3b). The lower temperature can suppress and slow down microbial growth in the superchilled fish. Low temperature helps to partially freeze most of the water in the fish muscle tissue [34]. This reduces water activity and makes the remaining free water hardly accessible to microorganisms. The metabolic processes occur at a much slower pace, which is the main reason for the low degree of bacterial microflora development [35]. The ultra-fast air-superchilling in our study has proven to be successful in all analyzed samples. The superchilling successfully inhibited the growth of microorganisms during 21 days of storage, thus preserving freshness and prolonging the shelf life of golden rainbow trout.

The data obtained from the microbiological analyses of all experimental samples are in full agreement with the results of the sensory analysis of the roasted golden rainbow trout and with the indicators TVB-N and biogenic amines. Similar to our results have been published by Erikson [36], reporting a shelf life of 20 days and a reduced total bacterial colony count in fillets of Atlantic salmon (*Salmo salar*).

All samples of grilled golden rainbow trout were evaluated with high complex scores after 21 days at superchilled storage. These results indicate that the sensory characteristics of fish were well preserved at all analyzed ultra-fast air regimes of superchilling, although sample 9 of golden rainbow trout received the highest estimated sensory score after grilling. Lower but very similar sensory scores were estimated for samples 4 and 8. The combination of sufficiently low temperatures and a high rate of ultra-fast air-superchilling allows for the production of superchilled golden rainbow trout with attractive sensory characteristics. The sensory scores evaluated for the grilled golden rainbow trout served as a parameter indicating fish freshness and could significantly affect the shelf life. The evaluated high sensory scores of grilled fish confirm that under superchilled conditions, it can be stored for up to 3 weeks after catching. Similar to ours, the results were reported by Cyprian et al. [37], who reached a 20-day shelf life of Nile tilapia (*Oreochromis niloticus*) fillet stored at $-1$ °C.

The air temperature most strongly affected the TVB-N formation in the golden rainbow trout muscle tissue. It was followed by the factor airflow rate (Figure 2a).

The right combination of these two factors (Figure 3a) in the superchilling process can provide a minimal rate of formation of TVB-N in the fish during its storage [35]. The air temperature of $-20$ °C and the airflow velocity of 3 m s$^{-1}$ could result in the formation of relatively large ice crystals inside the cells. They become the main cause of future denaturation of muscle proteins [38]. The established changes in sensory characteristics and microbiological parameters were in good agreement with the data obtained on the indicator of fish freshness—TVB-N.

According to obtained results presented in Figures 2b,c and 3b,c, the airflow velocity (υ) is most important for the TVB-N content, and the formation of TVC of microorganisms as the factor air temperature (T) is the second most influential factor. Similarly, Waterman and Taylor [39] supposed that the superchilling and storage of the Atlantic cod fillet at −1.1 °C extends the shelf life to about 20 days as the formation of ice crystals in the muscle tissue is negligible.

Decreasing the temperature to −2.2 °C keeps the fish at an acceptable freshness until 26 days, and the formation of ice crystals is more accelerated but does not exceed the acceptable limit. By decreasing the temperature to −2.8 °C, the shelf life is extended to 35 days, but damage caused by ice crystals makes the cod unfit for filleting.

The airflow velocity, along with the air temperature, plays a major role in optimizing the regime of superchilling from the aspect of the sensory characteristics of the grilled golden rainbow trout (Figure 2c).

The packaging material with a thickness of 80 μm retained fish freshness, preventing the tender skin of the golden rainbow trout from drying out under the influence of the relatively high cooling airflow velocity. At the same time, high cooling airflow velocity allows for a reduction in the period of time required to reach the desired temperature in the center of the fish. Therefore, shorter superchilling time preserves the structure of the fish muscle tissue, too. That is due to the formation of many in number and small in diameter ice crystals in fish musculature, which cannot harm and physically damage fish flesh [40]. The results of the sensory analysis are in agreement with the data obtained of the indicators, TVB-N, and biogenic amines, as well as with the TVC estimated in the superchilled golden rainbow trout. Our results are similar to those of Shen et al. [41], who studied rainbow trout fillets, and those of Duun and Rustad [42], who studied cod fillets (*Gadus morhua*) stored at −2.2 °C.

Different regimes of superchilling result in a different quality of golden rainbow trout after 21 days of storage from −2 to −3 °C. The results presented above (Figure 3) allow us to conclude that the change in each of the three analyzed variable factors affects differently the quality parameters of the fish. It was shown that the most significant influence of the variable factors on the sensory, physicochemical, and microbiological quality parameters turns out to be the air temperature (T), followed by the airflow velocity (υ).

The temperature intervals of superchilled storage are the main step to preserve muscle structure and minimize drip loss and muscle softening, which causes deterioration in the sensory characteristics of fish and accelerates the growth of surface microorganisms [43]. The process of ultra-fast air superchilling is very aggressive to the tender appearance of golden trout, so packaging, air velocity, and temperature play major roles in the deterioration of the initial qualitative characteristics, which lead to lower product quality during storage due to the more recently occurred proteolytic degradation. Small variations in temperature can cause large variations in the formation and size of ice crystals, which can lead to bigger liquid loss due to the freeze denaturation of muscle proteins [42].

Unlike freezing, superchilling does not reach temperatures below the cryoscopic freezing point and the crustal formation of water dipoles [35,36,38]. For this reason, the riskiest temperature interval (between −2 and −5 °C), where the most pronounced cold denaturation of muscle proteins is found, is never reached [6,18]. Muscle proteins largely retain their structure and functional biochemical characteristics [7,39]. That is why in the superchilled fish, no strongly expressed negative changes were found compared to the thawed fish, such as a significant deterioration of the sensory characteristics, an increased volume of droplet losses during deforestation, and an accelerated growth of surface microorganisms [6,14,16,18].

On the contrary, no significant denaturation changes in the muscle protein structures were found in the chilled fish, but on the other hand, the shelf life was approximately one week shorter due to the development of microbial spoilage [16,31].

The ultra-rapid air-superchilling process is very aggressive toward the relatively more delicate appearance of the golden rainbow trout compared to the normal one [26]. Therefore,

technological factors such as the thickness of the packaging material layer, air velocity, and air temperature play a major role in preserving the original quality characteristics and in minimizing the processes of proteolytic and lipolytic degradation of fish flash [38,41].

Ensuring an optimal regime of superchilling results in relatively small variations in the surface temperature of the fish. This is the reason why large variations in ice formation and ice crystal size are not found [33–35]. Thus, minimal drop loss is not guaranteed due to the negligible degree of cold denaturation of muscle proteins [7,32,42].

No presence of histamine was found (Figure 5) in the analyzed samples, which shows the relation between initial cells and storing of fish at lower temperatures to histamine formation levels [40,44].

## 5. Conclusions

The process of ultra-fast air-superchilling was applied to the golden rainbow trout (*Oncorchynchus mykiss*, Stevanovski) in order to optimize the quality of the produced fish (Figure 4). It was accomplished at the following process parameters: the air temperature T = −11.3 °C; the airflow velocity $v$ = 6.5 m s$^{-1}$; and the packing layer thickness D = 79.2 μm. Fish samples that undergone for superchilling at this regime had the lowest TVB-N content, the best possible sensory properties, and the minimal total viable count (TVC) of microorganisms, and in general, its freshness retention. Optimized results for the ultra-fast air superchilling process of golden rainbow trout can be used in the fish industry to improve power usage during the process and to obtain the best overall quality in production.

**Supplementary Materials:** The following supporting information can be downloaded at: https://www.mdpi.com/article/10.3390/app13179504/s1, Pictures of the apparatuses for vacuum-packing and superchilling.

**Author Contributions:** The working design of our work was developed with more than five days of working meetings in which all authors participated and gave full contributions based on their expertise. Realization of analyses was made by V.D.K., with the supervision of S.G.D., D.B.V.-V. and H.N.N. Data collection analyses and graphical visualization were performed by V.D.K., who also prepared the draft version of the article. The critical revision was performed by S.G.D., followed by D.K.B., D.B.V.-V. and H.N.N. Final approval of this manuscript was performed by S.G.D. All authors have read and agreed to the published version of the manuscript.

**Funding:** This study was supported by a Ph.D. students' funding body of the University of Food Technologies, Plovdiv, Bulgaria, and was supported by the company Intrama-Bulgaria Ltd., Dobrich, Bulgaria.

**Institutional Review Board Statement:** Not applicable.

**Informed Consent Statement:** Not applicable.

**Data Availability Statement:** Data available on request. The data presented in this study are available on request from the corresponding author.

**Acknowledgments:** The authors express their gratitude to the team of the Plovdiv office of company Intrama Bulgaria Ltd., Dobrich, for the provided consultancy assistance when choosing the material for preparing the individual vacuum packs, as well as technical and logistic support for their production and free donation. We thank them also for providing a single chamber vacuum packing machine.

**Conflicts of Interest:** We confirm that this manuscript has not been published elsewhere and is not under consideration by another journal. All authors have approved the manuscript and agree with submission to the Journal of Applied Sciences. The authors have no conflict of interest to declare.

## Abbreviations

CCD—central composite design; D—packaging layer thickness; HCBB-BC—hybrid chaotic big bang-big crunch mapping algorithm; HPLC—high-performance liquid chromatography; NMR—

nuclear magnetic resonance; P—electric current power; PVDC—polyvinyl dichloride; T—air temperature of superchilling; TVB-N—total nitrogen volatile bases; TVC—total number of microorganisms; υ—airflow velocity of superchilling; 3D NoC—three-dimensional network-on-chip.

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
