# Peer review of "Nutritional Quality and Overall Acceptability Optimization of Ultra-Fast Air-Superchilled Golden Rainbow Trout (Oncorhynchus mykiss, Stevanovski) Using the Response Surface Methodology"

_applsci, doi:10.3390/app13179504_

Round 1

Reviewer 1 Report

Paper entitled “Optimization of the Process of Ultra-fast Air Super chilling of Golden Rainbow Trout (Oncorhynchus mykiss, Stevanovski)”

-          Title of paper: In optimization we must optimize the studied response (Nutritional Quality of the studied matter - Golden Rainbow Trout (Oncorhynchus mykiss, Stevanovski), in the present paper), not the used process applied to obtain the response. For this reason, we propose the following title instead of the old one: “Nutritional Quality and Overall Acceptability Optimization of Ultra-Fast Air Super Chilled Golden Rainbow Trout (Oncorhynchus mykiss, Stevanovski) using the Response Surface Methodology”.

-          Abstract, Lines 3-4: For the same reason, please replace the following sentence: “The aim of this study, was to optimize the regime of ultra-fast air super chilling of golden rainbow trout.” by “The aim of this study, was to optimize the nutritional quality and the overall acceptability (in term of Total Volatile Base Nitrogen (TVB-N), Total Viable Counts (TVC), and Sensory scores) of ultra-fast air super chilled golden rainbow trout.”

-          Abstract, Lines 8-11: For the same reason, please replace the following sentence: “According to the obtained values for TVBN, TVC and sensory scores, an optimal regime of ultra-fast air super chilling of the golden rainbow trout has been found at following process parameters: air temperature T= -11.3℃, airflow velocity υ=6.5 m s-1 and packaging layer thickness D=79.2 μ.” by “According to the optimized values for TVB-N, TVC and sensory scores, giving us better quality of ultra-fast air super chilled golden rainbow, the process regime has been found at following parameters: air temperature T= -11.3℃, airflow velocity υ=6.5 m s-1 and packaging layer thickness D=79.2 μ.”

-          Throughout the paper: Please verify if “Stevanovski” must be in italic format or not.

-          Abstract, Line 9: Please replace “TVBN” by “TVB-N”.

-          Introduction: Please replace the last sentence: “That is why the objective of this study was to optimize the regime of golden rainbow trout ultra-fast air super chilling.” by “That is why the objective of this study was to determine the best process regime of ultra-fast air super chilling, giving us the optimal quality of golden rainbow trout”.

-          Table 1 title: Please replace “experiment with three factors at two levels (32+ additional experiments in the center)” by “experiment with three factors at five levels designed by using Central Composite Design (CCD) with a = 1.68179”; here we talk about 5 levels (-a; -1; 0; +1; +a) not 2. Please verify the used value of a. Please also see books/papers talking about the Central Composite Design (CCD).

-          Just before Table 1: Please define the studied factors, their actual levels, and their reduced centered levels (Xi). Also, authors should present the goal of this study here.

-          Table 1: Please replace:

o   “(D, μ)” by “(D, μm)” in first line; last column. Because we talk about thickness.   

o   “- -“ by “- a” in experiment 9 ; column 2.

o   “+ +“ by “+ a” in experiment 10 ; column 2.

o   “- -“ by “- a” in experiment 11 ; column 3.

o   “+ +“ by “+ a” in experiment 12 ; column 3.

o   “- -“ by “- a” in experiment 13 ; column 4.

o   “+ +“ by “+ a” in experiment 14 ; column 4.

-          Last sentence of section “2.2 Auxiliary materials”, Please replace “100 μ” by “100 μm” because we talk about thickness.  

-          Line 2 of Paragraph 1 of section “2.3 Exploratory factor analyses design”, “Oncorhynchus mykiss” must be in italic format.

-          Line 3 of Paragraph 1 of section “2.3 Exploratory factor analyses design”, Please replace “two” by “five”.

-          Line 5 of Paragraph 1 of section “2.3 Exploratory factor analyses design”, Please replace “(+ +)” by “(+ a)”.

-          Line 6 of Paragraph 1 of section “2.3 Exploratory factor analyses design”, Please replace “(- -)” by “(- a)”.

-          Line 7 of Paragraph 1 of section “2.3 Exploratory factor analyses design”, Please remove “(32 + additional trials in the center)”. It is a false information.

-          Line 3 of Paragraph 2 of section “2.3 Exploratory factor analyses design”, Please replace “(D, μ)” by “(D, μm)”. Because we talk about thickness.  

-          Table 2 title: Please replace “design (32)” by “Central Composite Design”.

-          Last line of section “2.7 Determination of biogenic amines”: Please add “,” after “(10 kg)”.

-          First Line of section “2.8 Statistical analyses”: Please replace the following sentence “For optimization of the process of ultra-fast air super chilling the method of exploratory factor analysis [16]” by “For optimization of the quality of the ultra-fast air super chilled studied fish, the method of exploratory factor analysis [16]”.

-          Second line of section “2.8 Statistical analyses”: Please remove “(32 + additional trials in the center)”.

-          Third line of section “2.8 Statistical analyses”: Please replace “For the final purpose – optimization of the process, desirability test was made and the visualization was presented with desirability plot.” by “For this final purpose, desirability test was made and the visualization was presented with desirability plot.”

-          Table 3:

o   Column 4; second line: Please replace “μ” by “μm”.

o   Column 5; last line: Please replace “23,23” by “23.23”.

-          Table 3: is not adequate with the presented table 1 in section “2.1 Fish samples labelling” of Materials and Methods part. In this case the obtained results can’t be solid and we cannot consider them for optimization. Please add the results of the remaining experiments. The discussion must be also updated after this modification.

-          Third line of Figure 1 title: Please replace “(a) Imact on TVB-N content” by “(a) Impact on TVB-N content”.

-          Figure 2 title: In fact, here the presented figure presents only the desirability as function of two factors (Temperature and Air flow velocity), not of three factors. For this, please replace in the first line of the title “three examined variables: the air temperature (T), the airflow velocity (υ) and the packing layer thickness (D)” by “two examined variables: the air temperature (T) and the airflow velocity (υ)”.

-          Figure 2 axis:

o   Please replace the title of the first axis “Temperature” by “Temperature (°C)”.

o   Please replace the title of the second axis “Air Flow” by “Air flow velocity (m s-1)”.

o   Please add the title of the third axis “Desirability (-)”.

-          The quality of figure 3 is very poor and it is not clear. Please replace it with another that can be better.

-          Part “4. Discussion”:

o   Paragraph 1; line 2: with adding of the remaining experiments, replace “9” by “17”.

o   Paragraph 3; lines 4, 12 and 15: with adding of the remaining experiments, replace “nine” by “seventeen”.

o   Paragraph 3; line 5: Please add one space between “7.00 log cfu g-1” and “[14]”.

o   Paragraph 4; lines 1 and 3: with adding of the remaining experiments, replace “nine” by “seventeen”.

o   Paragraph 7; line 4: with adding of the remaining experiments, please verify “Thay”.

o   Paragraph 10; line 1: Please replace “μ” by “μm”.

-          Part “5. Conclusion”:

o   Line 1: Please replace the following sentence “An optimal ultra-fast air super chilling regime of the golden rainbow trout (Oncorchynchus mykiss, Stevanovski) has been established.” by “The process of ultra-fast air super chilling was applied to the golden rainbow trout (Oncorchynchus mykiss, Stevanovski) in order to optimize the quality of the produced fish”.

o   Line 4: Please replace “D = 79.2 μ” by “D = 79.2 μm”.

Author Response

Manuscript ID: applsci-2565023

Dear Reviewer 1,

First of all, please accept our gratitude for the critical notes and constructive comments aimed at improving the quality, informativeness and presentability of our manuscript entitled "Optimization of the Process of Ultra-fast Air Super chilling of Golden Rainbow Trout (Oncorhynchus mykiss, Stevanovski)".

The authors' response to the reviewers' recommendations

Note: Changes made in accordance with the recommendations of the reviewers are highlighted in yellow.

-    Throughout the paper: Please verify if “Stevanovski” must be in italic format or not.

Authors' response: We have very carefully and thoroughly studied how it is correct to write the systematic name of the fish. In all sources we encountered italics for the Latin name of the fish and regular font - for the name of the discoverer and the year. For example, about normal standard rainbow trout (Oncorhynchus mykiss Walbaum, 1792) and about golden rainbow trout (Oncorhynchus mykiss Stevanovski, 1987).  That is why throughout the article we leave the original spelling of the Latin name of the fish species. (Oncorhynchus mykiss, Stevanovski)".

-    Title of paper: In optimization we must optimize the studied response (Nutritional Quality of the studied matter - Golden Rainbow Trout (Oncorhynchus mykiss, Stevanovski), in the present paper), not the used process applied to obtain the response. For this reason, we propose the following title instead of the old one: “Nutritional Quality and Overall Acceptability Optimization of Ultra-Fast Air Super Chilled Golden Rainbow Trout (Oncorhynchus mykiss, Stevanovski) using the Response Surface Methodology”.

Authors' response: We accept the suggestion made by the reviewer as reasonable and have changed the title of the article. Thanks for the constructive and precise suggestion.

- Abstract

-    Abstract, Lines 3-4: For the same reason, please replace the following sentence: “The aim of this study, was to optimize the regime of ultra-fast air super chilling of golden rainbow trout.” by “The aim of this study, was to optimize the nutritional quality and the overall acceptability (in term of Total Volatile Base Nitrogen (TVB-N), Total Viable Counts (TVC), and Sensory scores) of ultra-fast air super chilled golden rainbow trout.”

-    Abstract, Lines 8-11: For the same reason, please replace the following sentence: “According to the obtained values for TVBN, TVC and sensory scores, an optimal regime of ultra-fast air super chilling of the golden rainbow trout has been found at following process parameters: air temperature T= -11.3℃, airflow velocity υ=6.5 m s-1 and packaging layer thickness D=79.2 μ.” by “According to the optimized values for TVB-N, TVC and sensory scores, giving us better quality of ultra-fast air super chilled golden rainbow, the process regime has been found at following parameters: air temperature T= -11.3℃, airflow velocity υ=6.5 m s-1 and packaging layer thickness D=79.2 μ.”

-    Abstract, Line 9: Please replace “TVBN” by “TVB-N”.

Authors' response: We accept the suggestions made by the reviewer and have changed the aim of the study and next two suggestions. Thanks for the constructive and precise suggestions.

-    Introduction:

Please replace the last sentence: “That is why the objective of this study was to optimize the regime of golden rainbow trout ultra-fast air super chilling.” by “That is why the objective of this study was to determine the best process regime of ultra-fast air super chilling, giving us the optimal quality of golden rainbow trout”.

Authors' response: We accept the suggestions made by the reviewer and have changed the last sentence of the Introduction section. Thanks for the constructive suggestion.

- Table 1:

-    Table 1 title: Please replace “experiment with three factors at two levels (32+ additional experiments in the center)” by “experiment with three factors at five levels designed by using Central Composite Design (CCD) with a = 1.68179”; here we talk about 5 levels (-a; -1; 0; +1; +a) not 2. Please verify the used value of a. Please also see books/papers talking about the Central Composite Design (CCD).

-    Just before Table 1: Please define the studied factors, their actual levels, and their reduced centered levels (Xi). Also, authors should present the goal of this study here.

-    Table 1: Please replace:

o     “(D, μ)” by “(D, μm)” in first line; last column. Because we talk about thickness.  

o     “- -“ by “- a” in experiment 9 ; column 2.

o     “+ +“ by “+ a” in experiment 10 ; column 2.

o     “- -“ by “- a” in experiment 11 ; column 3.

o     “+ +“ by “+ a” in experiment 12 ; column 3.

o     “- -“ by “- a” in experiment 13 ; column 4.

o     “+ +“ by “+ a” in experiment 14 ; column 4.

Authors' response: We accept the suggestions made by the reviewer according Table 1 and have changed it. Thanks for the constructive suggestions and criticism.

-    Materials and Methods:

-    Last sentence of section “2.2 Auxiliary materials”, Please replace “100 μ” by “100 μm” because we talk about thickness.  

-    Line 2 of Paragraph 1 of section “2.3 Exploratory factor analyses design”, “Oncorhynchus mykiss” must be in italic format.

-    Line 3 of Paragraph 1 of section “2.3 Exploratory factor analyses design”, Please replace “two” by “five”.

-    Line 5 of Paragraph 1 of section “2.3 Exploratory factor analyses design”, Please replace “(+ +)” by “(+ a)”.

-    Line 6 of Paragraph 1 of section “2.3 Exploratory factor analyses design”, Please replace “(- -)” by “(- a)”.

-    Line 7 of Paragraph 1 of section “2.3 Exploratory factor analyses design”, Please remove “(32 + additional trials in the center)”. It is a false information.

-    Line 3 of Paragraph 2 of section “2.3 Exploratory factor analyses design”, Please replace “(D, μ)” by “(D, μm)”. Because we talk about thickness.  

-    Table 2 title: Please replace “design (32)” by “Central Composite Design”.

-    Last line of section “2.7 Determination of biogenic amines”: Please add “,” after “(10 kg)”.

-    First Line of section “2.8 Statistical analyses”: Please replace the following sentence “For optimization of the process of ultra-fast air super chilling the method of exploratory factor analysis [16]” by “For optimization of the quality of the ultra-fast air super chilled studied fish, the method of exploratory factor analysis [16]”.

-    Second line of section “2.8 Statistical analyses”: Please remove “(32 + additional trials in the center)”.

-    Third line of section “2.8 Statistical analyses”: Please replace “For the final purpose – optimization of the process, desirability test was made and the visualization was presented with desirability plot.” by “For this final purpose, desirability test was made and the visualization was presented with desirability plot.”

Authors' response: We accept all critical notes of the reviewer and have changed every one of them. Thanks for the constructive proposals.

-    Table 3:

o   Column 4; second line: Please replace “μ” by “μm”.

o   Column 5; last line: Please replace “23,23” by “23.23”.

-    Table 3: is not adequate with the presented table 1 in section “2.1 Fish samples labelling” of Materials and Methods part. In this case the obtained results can’t be solid and we cannot consider them for optimization. Please add the results of the remaining experiments. The discussion must be also updated after this modification.

Authors' response: We accept the critical notes of the reviewer about Table 3 and have changed every one of them. Thank you for the remarks.

-    Figure 1

Third line of Figure 1 title: Please replace “(a) Imact on TVB-N content” by “(a) Impact on TVB-N content”.

Authors' response: We accept the critical note of the reviewer about Figure 1 and have changed this figure. Thank you for the comments.

-   Figure 2

-    Figure 2 title: In fact, here the presented figure presents only the desirability as function of two factors (Temperature and Air flow velocity), not of three factors. For this, please replace in the first line of the title “three examined variables: the air temperature (T), the airflow velocity (υ) and the packing layer thickness (D)” by “two examined variables: the air temperature (T) and the airflow velocity (υ)”.

-    Figure 2 axis:

o   Please replace the title of the first axis “Temperature” by “Temperature (°C)”.

o   Please replace the title of the second axis “Air Flow” by “Air flow velocity (m s-1)”.

o   Please add the title of the third axis “Desirability (-)”.

Authors' response: We accept the critical notes of the reviewer about Figure 2 and have changed it. Thank you for the remarks  and comments.

-   Figure 3

-    The quality of figure 3 is very poor and it is not clear. Please replace it with another that can be better.

Authors' response: We accept the critical notes of the reviewer about Figure 3 /in revised version of the manuscript – Figure 6/ and have changed it. Thank you for the comments.

-    Part “4. Discussion”:

o   Paragraph 1; line 2: with adding of the remaining experiments, replace “9” by “17”.

o   Paragraph 3; lines 4, 12 and 15: with adding of the remaining experiments, replace “nine” by “seventeen”.

o   Paragraph 3; line 5: Please add one space between “7.00 log cfu g-1” and “[14]”.

o   Paragraph 4; lines 1 and 3: with adding of the remaining experiments, replace “nine” by “seventeen”.

o   Paragraph 7; line 4: with adding of the remaining experiments, please verify “Thay”.

o   Paragraph 10; line 1: Please replace “μ” by “μm”.

Authors' response: We accept the critical notes of the reviewer about section Discussion. We have changed majority of them.  In some cases, we delated “nine” only to avoid redundant diphthongs and because it is implied from the preceding text. Thank you for understanding.

-    Part “5. Conclusion”:

o   Line 1: Please replace the following sentence “An optimal ultra-fast air super chilling regime of the golden rainbow trout (Oncorchynchus mykiss, Stevanovski) has been established.” by “The process of ultra-fast air super chilling was applied to the golden rainbow trout (Oncorchynchus mykiss, Stevanovski) in order to optimize the quality of the produced fish”.

o   Line 4: Please replace “D = 79.2 μ” by “D = 79.2 μm”.

Authors' response: We accept the proposal and critical notes of the reviewer about section Conclusion. We have changed the specified texts. Thank you for your propositions.

Reviewer 2 Report

Please read the attachment. Thank you.

Author Response

Manuscript ID: applsci-2565023

Dear Reviewer 2,

First of all, please accept our gratitude for the critical notes and constructive comments aimed at improving the quality, informativeness and presentability of our manuscript entitled "Optimization of the Process of Ultra-fast Air Super chilling of Golden Rainbow Trout (Oncorhynchus mykiss, Stevanovski)".

The authors' response to the reviewers' comments and suggestions

Note: Changes made in accordance with the recommendations of the reviewers are highlighted in yellow.

Regarding Specific Comments we provide our opinions below:

-    Keywords: Please provide between 5 and 10 keywords. Which are not repeated words or phrases in the title.

Authors' response: We accept the remark and consider it well-founded. In the revised version of the manuscript, the keywords were generally changed and enriched.

-    Introduction: please add a paragraph to introduce the outline of the manuscript.

Authors' response: We accept the remark. We have added three new paragraphs that clarify our outline of the manuscript.

-    Section 3: “Results” should be “Results and Discussion.”

Authors' response: We reject the suggestion made by the reviewers. According to this comment, we would like to note that when drafting the manuscript, we strictly follow the manuscript preparation template provided by the Applied Science Instructions for Authors, where the Results and Discussion sections are presented as separate sections. For this reason, we consider it appropriate to adhere to the journal style.

-    Figure 3: its quality is too blurry. Please increase its resolution.

Authors' response: We accept the remark. The resolution of Figure 3 has been increased, which after adding two more figures (according to the next proposal) became Figure 6.

-    Supplementary Materials: Please add all figures to the main text because this work seems to have insufficient results.

Authors' response:   We accept the remark. We add two more figures to the main text from the analyzes made in our work.

-    All results were not explained and analyzed enough. Please revise.

Authors' response: We accept the remark. We enrich the Discussion section and more precisely introduce several new references and explain in detail about our findings and the related work already done in the literature review.

-    Related work: this section is too short. Please extend this section. Please add some missing literature reviews. The related work could be considered as the applications of the algorithms in the practical industry. The following work could be helpful and enrich this specific section. 

+ Optimization of Meat and Poultry Farm Inventory Stock Using Data Analytics for Green Supply Chain Network

+ Optimizing magnification ratio for the flexible hinge displacement amplifier mechanism design

Authors' response: We accept the remark. We included the first one (8.Kler, R.; Gangurde, R.; Elmirzaev, S.; Hossain, S.; Vo, N.; Nguyen, T.; Kumar, N. Optimization of Meat and Poultry Farm Inventory Stock Using Data Analytics for Green Supply Chain Network, Discrete Dyn. Nat. Soc. 2022, 1(1), 1-8. https://doi.org/10.1155/2022/8970549) under number 8 in the list of References and  used the information to quality and informativeness of the Introduction session.  This section has been expanded. Added 14 new literature sources. Related work has also addressed applications of the algorithms in practical industry.

-    Please provide a flowchart of the optimization process.

Authors' response: We accept the remark. We add a flowchart of the optimization process in the Materials and Methods section (Figure 1).

-    Conclusion: The conclusion should concisely summarize the study's key findings and their implications for the fish industry. The authors should avoid overgeneralized statements and focus on the specific results and their practical applications.

Authors' response: We accept the remark. We improve and make strong and concise conclusions which are directly related to the practical applications in fish industry and advantages at comprehensive approach related with all stakeholders in the ecosystem.

-    References are short (only 31 refs). Please improve and enrich your references.

Authors' response:  We accept the remark. We improve and enrich our references and from 31 now they are  44.

Answers of the Reviewer's Questions

We want to thank of the reviewers to the Constructive Questions, they helped our manuscript to became more concise and stronger.

Our comments are as follows:

  1. How does the proposed ultra-fast air super chilling regime compare to conventional chilling methods to preserve the quality and freshness of golden rainbow trout? Are any significant differences observed in sensory analysis, TVB-N, TVC, and biogenic amines between the optimized super chilled and conventionally chilled fish?

Authors' response:  We have conducted research on these issues. We have published the results in the scientific article "Extend the shelf life of fresh golden rainbow trout by ultra-rapid super chilling with air or cryogenic carbon dioxide“ by Kitanovski V. D., D. B. Vlahova-Vangelova, S. G. Dragoev, H. N. Nikolov, D. K. Balev, published in the Journal of Aquaculture Research & Development. 2017, 8(4): 100481. https://doi.org/10.4172/2155-9546.1000481

  1. Considering the potential variations in fish size and initial microbial loads, did the authors investigate the correlation between fish size, microbial activity, and the effectiveness of the optimized super chilling regime? Were there any specific size categories that exhibited enhanced results?

Authors' response:  The size of the fish used in the experiments was relatively constant. This is due to the fact that the fish is supplied from a trout aquaculture farm. The fish were hatched at the same time from a uniform batch of spawning material. It is grown and fed under the same conditions. The fish age is also the same. In this sense, all caught and examined fish have a relatively uniform own weight of 370±10 g each. For these reasons, the "size" and/or "mass" parameter of the fish is relatively constant. Therefore, it is not specifically investigated, but is taken as a constant. The relationship between subcooling regimes and microbial activity was investigated. The data are published in Table 3 of the above-cited publication (Kitanovski V. D., D. B. Vlahova-Vangelova, S. G. Dragoev, H. N. Nikolov, D. K. Balev. Extend the shelf life of fresh golden rainbow trout by ultra-rapid super chilling with air or cryogenic carbon dioxide. Journal of Aquaculture Research & Development, 2017, 8(4): 100481. https://doi.org/10.4172/2155-9546.1000481), as well as in Table. 3 of this manuscript. Data are presented on the influence of optimized super chilling regimes on the total viable count of microorganisms.

  1. In the discussion section, could the authors elaborate on the underlying mechanisms behind the observed proteolytic degradation delay and the reduced microbial activity in the super-chilled fish? Are there any plausible biochemical or biophysical explanations for these effects at the proposed air temperature, airflow velocity, and packaging layer thickness?

Authors' response:  The storage temperature ranges of super chilled products, when the surface of the fish reaches a temperature characteristic of a super chilled product, is the main reason for preserving muscle structure and minimizing drip loss and muscle softening. Unlike freezing, super chilling does not reach temperatures below the cryoscopic freezing point and crustal formation of water dipoles. For this reason, the riskiest temperature interval (between -2 and -5℃), where the most pronounced cold denaturation of muscle proteins is found, is never reached. Muscle proteins largely retain their structure and functional biochemical characteristics. That is why in the super chilled fish no strongly expressed negative changes were found comparing to the thawed fish, such as a significant deterioration of the sensory characteristics, an increased volume of droplet losses during deforestation and an accelerated growth of surface microorganisms.

On the contrary, no significant denaturation changes of the muscle protein structures were found in the chilled fish, but on the other hand, the shelf life was approximately one week shorter due to the development of microbial spoilage.

The ultra-rapid air super chilling process is very aggressive towards the relatively more delicate appearance of the golden rainbow trout compared to the normal one. Therefore, technological factors such as: thickness of the packaging material layer, air velocity and air temperature play a major role in preserving the original quality characteristics and in minimization the processes of proteolytic and lipolytic degradation of fish flash.

Ensuring an optimal regime of super chilling results in relatively small variations in the surface temperature of the fish. This is the reason why large variations in ice formation and ice crystal size are not found. Thus, minimal drop loss is not guaranteed due to the negligible degree of cold denaturation of muscle proteins.

Reviewer 3 Report

The aim of this manuscript is to optimize the regime of ultra-fast air super chilling of golden rainbow trout.

This manuscript shows rich content, providing a deep insight for some works: the study is within the journal’s scope, and I found it to be well-written, providing sufficient information. Even if the manuscript provides an organic overview, with a densely organized structure and based on well-synthetized evidence, there are some suggestions necessary to make the article complete and fully readable. For these reasons, the manuscript requires major changes.

Please find below an enumerated list of comments on my review of the manuscript:

ABSTRACT:

The authors should rewrite this sentence, as following: “The aim of this study was to optimize the regime of ultra-fast air super chilling of golden rainbow trout”.

The authors should provide a list of the abbreviations, mentioned in the manuscript.

INTRODUCTION:

Specifically, in albinism genes such as Tyr-1, Tyr-2 and slc45a2 are associated to pigmentation (see, for reference: https://doi.org/10.1371/journal.pone.0214034). The manuscript may benefit from reporting the most significant genetic markers, involved in albinism of the rainbow trout.

Regarding the effects of temperature, recent evidence also suggested that temperature also exerts a pivotal role on the development and resolution of pathologies in rainbow trout (see, for reference: https://doi.org/10.1016/j.aquaculture.2022.738577), which might compromise the shelf life of the fish. The authors should highlight this issue, which have a considerable impact on the survival of the fish.

The main topic is interesting, and certainly of great clinical impact. As regards the originality and strengths of this manuscript, this is a significant contribute to the ongoing research on this topic, as it extends the research field on the optimization of the regime of ultra-fast air super chilling of golden rainbow trout. Overall, the contents are rich, and the authors also give their deep insight for some works.

As regards the section of methods, there is a specific and detailed explanation for the methods used in this study: this is particularly significant, since the manuscript relies on a multitude of methodological and statistical analysis, to derive its conclusions. The methodology applied is overall correct, the results are reliable and adequately discussed.

The conclusion of this manuscript is perfectly in line with the main purpose of the paper: the authors have designed and conducted the study properly. As regards the conclusions, they are well written and present an adequate balance between the description of previous findings and the results presented by the authors.

In conclusion, this manuscript is densely presented and well organized, based on well-synthetized evidence. The authors were lucid in their style of writing, making it easy to read and understand the message, portrayed in the manuscript. Besides, the methodology design was appropriately implemented within the study. However, many of the topics are very concisely covered. This manuscript provided a comprehensive analysis of current knowledge in this field. Moreover, this research has futuristic importance and could be potential for future research. However, major concerns of this manuscript are with the introductive section: for these reasons, I have major comments for this section, for improvement before acceptance for publication. The article is accurate and provides relevant information on the topic and I have some major points to make, that may help to improve the quality of the current manuscript and maximize its scientific impact. I would accept this manuscript if the comments are addressed properly.

Author Response

Manuscript ID: applsci-2565023

Dear Reviewer 3,

First of all, please accept our gratitude for the critical notes and constructive comments aimed at improving the quality, informativeness and presentability of our manuscript entitled "Optimization of the Process of Ultra-fast Air Super chilling of Golden Rainbow Trout (Oncorhynchus mykiss, Stevanovski)".

Thank you for appreciating our work and our proposed manuscript so highly.

We accept all your suggestions for major changes to the manuscript as follows:

The authors' response to the reviewers' comments and suggestions

Note: Changes made in accordance with the recommendations of the reviewers are highlighted in yellow.

-    Abstract: The authors should rewrite this sentence, as following: “The aim of this study was to optimize the regime of ultra-fast air super chilling of golden rainbow trout”.

Authors' response: We accept the remark and consider it well-founded. In the revised version of the manuscript, the aim of the study in the Abstract and in the Introduction was generally preformulated. In addition, we change the Title of the manuscript too.

-    Abbreviations: The authors should provide a list of the abbreviations, mentioned in the manuscript.

Authors' response: We accept the remarks of the reviewer. In the revised version of the manuscript, the keywords were generally changed and enriched and abrreviations used were added:

Keywords: fish; freshness; shelf life; coolant velocity; coolant temperature; package thickness; total volatile base nitrogen (TVB-N); total number of mesophilic microorganisms (TVC); histamine; sensory properties

Abbreviations: CCD - central composite design; D - packaging layer thickness; HCBB-BC - hybrid chaotic big bang-big crunch mapping algorithm; HPLC - high-performance liquid chromatography; NMR - nuclear magnetic resonance; P - electric current power; PVDC - polyvinyl dichloride; T - air temperature of super chilling; TVB-N - total nitrogen volatile bases; TVC - total number of microorganisms; υ - airflow velocity of super chilling; 3D NoC - three-dimensional network-on-chip  

-    Introduction: Specifically, in albinism genes such as Tyr-1, Tyr-2 and slc45a2 are associated to pigmentation (see, for reference: https://doi.org/10.1371/journal.pone.0214034). The manuscript may benefit from reporting the most significant genetic markers, involved in albinism of the rainbow trout.

Regarding the effects of temperature, recent evidence also suggested that temperature also exerts a pivotal role on the development and resolution of pathologies in rainbow trout (see, for reference: https://doi.org/10.1016/j.aquaculture.2022.738577), which might compromise the shelf life of the fish. The authors should highlight this issue, which have a considerable impact on the survival of the fish.

However, major concerns of this manuscript are with the introductive section: for these reasons, I have major comments for this section, for improvement before acceptance for publication.

Authors' response: We accept the remark and considerations of the reviewer. In the revised version of the manuscript, the first two cited authors were replaced with two new sources shown by the Reviewer 3. Thank you for the constructive propositions. The Introduction part was extended approximately twice.

Other parts of the manuscript, as well as the References that are cited, also underwent substantial changes.

Round 2

Reviewer 3 Report

The authors have improved this manuscript in an organic way. I accept for the publication.